# Nonlinear spin-orbit coupling in optical thin films

**Domenico de Ceglia** [1,2,7] ✉, **Laure Coudrat** [3,7] ✉, **Iännis Roland**[3],
**Maria Antonietta Vincenti**[1,2], **Michael Scalora**[4], **Rana Tanos**[5], **Julien Claudon** [5],
**Jean-Michel Gérard** [5], **Aloyse Degiron** [3], **Giuseppe Leo**[3,6] &
**Costantino De Angelis** [1,2]

Tunable generation of vortex beams holds relevance in various fields, including communications and sensing. In this paper, we demonstrate the feasibility of nonlinear spin-orbit interactions in thin films of materials with second-order nonlinear susceptibility. Remarkably, the nonlinear tensor can mix the longitudinal and transverse components of the pump field. We observe experimentally our theoretical predictions in the process of second-harmonic generation from a thin film of aluminum gallium arsenide, a material platform widely spread for its role in the advancement of active, nonlinear, and quantum photonic devices. In particular, we prove that a nonlinear thin film can be used to produce vector vortex beams of second-harmonic light when excited by circularly-polarized Gaussian beams.

Spin-orbit interactions[1] have been observed under different circumstances in linear optics: (i) conversion of spin angular momentum (SAM) to orbital angular momentum (OAM) in tightly focused beams propagating in isotropic and homogeneous media[2], (ii) spin Hall effect at dielectric interfaces illuminated with circularly-polarized light[3], (iii) spin-orbit interactions for paraxial beams propagating along the optical axis of uniaxial homogeneous crystals[4] or anisotropic and inhomogeneous structures[5], (iv) quantum spin Hall effect of light due to spin-orbit interactions at metal-dielectric interfaces[6,7]. These interactions are enhanced in the presence of structured light and structured media (metamaterials, metasurfaces, and any kind of nanostructures) due to the presence of large inhomogeneities of near fields[8].

Spin-orbit interactions have been also investigated in nonlinear optics. Arbitrary charge addition, besides the intuitive charge doubling under phase-matching conditions[9], has been achieved in polarization-controlled second-harmonic generation (SHG), by combining orthogonally polarized pump beams carrying OAM with different topological charges[10,11]. SAM to OAM conversion has been observed in a barium borate (BBO) crystal pumped with a circularly polarized beam, in which the interplay between linear spin-orbit coupling and SHG is exploited to generate a variety of OAM states[12]. Interaction between OAM and SAM of light has been theoretically investigated in three-wave mixing processes occurring in bulk isotropic chiral media[13,14]. Spin-orbit angular momentum transfer has been also observed in noncollinear SHG from potassium titanyl phosphate (KTP) crystal under type-II phase matching, where the two interacting pump components are a vector vortex beam and a circularly polarized Gaussian beam[15]. Recently, a novel mechanism for optical angular momentum transfer to nonabsorbing nanoparticles has been proposed, in which an optical torque is induced via harmonic generation[16]. The study of light-matter interactions mediated by spin-orbit coupling is improving our understanding of the fundamentals of vectorial nonlinear optics, with implications in a variety of applications, including quantum information processing[17] and imaging[18].

Here we report theoretical and experimental evidence of a new type of nonlinear spin-orbit interaction mediated by the longitudinal field of a circularly polarized Gaussian beam, in which SAM to OAM

[1]CNIT and Department of Information Engineering, University of Brescia, Via Branze, 38, Brescia 25123, Italy. [2]Istituto Nazionale di Ottica, Consiglio Nazionale delle Ricerche, Via Branze, 45, Brescia 25123, Italy. [3]Laboratoire Matériaux et Phénomènes Quantiques, Université Paris Cité, CNRS, 10 rue A. Domon et L. Duquet, Paris 75013, France. [4]Charles M. Bowden Research Center, Redstone Arsenal, AL 35898-5000, USA. [5]Univ. Grenoble Alpes, CEA, Grenoble INP, IRIG, PHELIQS, "Nanophysique et Semiconducteurs" Group, Grenoble F-38000, France. [6]Institut universitaire de France (IUF), Paris, France. [7]These authors contributed equally: Domenico de Ceglia, Laure Coudrat. ✉e-mail: domenico.deceglia@unibs.it; laure.coudrat@u-paris.fr

transfer is achieved via SHG. Its occurrence requires that at least one of the interacting fields of the three-wave mixing process be the longitudinal field component of the pump beam. The effect can be induced by circularly polarized light interacting with media with a bulk second-order nonlinear susceptibility $\chi^{(2)}_{ijk}$ in which at least one of the two subscripts $j$ and $k$ is equal to $z$, i.e., the longitudinal coordinate. One of these crystals is aluminum gallium arsenide (AlGaAs), which is characterized by a quadratic susceptibility of type $\chi^{(2)}_{ijk}$ with $i \neq j \neq k$. This SAM to OAM transfer does not require phase matching between the pump and the second harmonic (SH) fields, and can be therefore observed in SHG from a thin film of (001)-cut AlGaAs.

We first present the theoretical background of spin-orbit coupling involving the longitudinal field, both in the linear and nonlinear regimes. By showing the peculiar manifestation of this spin-orbit coupling in AlGaAs with pumps carrying arbitrary combinations of SAM and OAM, we provide the rules that determine the SAM and OAM of the SH vortex beams generated in reflection and transmission. Then we report experimental results of SHG from a thin film of (001) AlGaAs on a transparent substrate excited with pump beams carrying different combinations of SAM and OAM, showing spin-orbit coupling effects in full agreement with our theoretical predictions. It is quite remarkable that this effect has not been observed until now, although it occurs in a semiconductor that is largely employed in electronics and photonics.

## Results
### Longitudinal field in a Gaussian beam carrying SAM

The amplitude of the transverse field in a Gaussian beam at its waist ($z = 0$) can be expressed as $E_0 = A e^{-\frac{x^2+y^2}{w^2}}$, where $w$ is the beam waist and $A$ the peak amplitude. The transverse component of the electric field vector at the $z = 0$ plane and at the frequency $\omega$ reads as:

$$\mathbf{E}^{\omega}_{\perp} = (\alpha \hat{x} + j\beta \hat{y})E_0 \tag{1}$$

where $j = \sqrt{-1}$, and $\alpha$ and $\beta$ are free (complex) parameters that define the polarization state of the beam $\hat{e} = \alpha \hat{x} + j\beta \hat{y}$. In Fourier space, one may write, $\tilde{\mathbf{E}}^{\omega}_{\perp} = (\alpha \hat{x} + j\beta \hat{y})\tilde{E}_0$.

Maxwell's equations in the absence of charges impose that $\nabla \cdot \mathbf{E}^{\omega} = 0$. Therefore $\mathbf{k} \cdot \tilde{\mathbf{E}}^{\omega} = 0$, with $\mathbf{k}$ indicating the wavevector, $\mathbf{E}^{\omega} = \mathbf{E}^{\omega}_{\perp} + \hat{z}E^{\omega}_z$ the electric field vector with Cartesian components $(E^{\omega}_x, E^{\omega}_y, E^{\omega}_z)$, and $\tilde{\mathbf{E}}^{\omega} = \tilde{\mathbf{E}}^{\omega}_{\perp} + \hat{z}\tilde{E}^{\omega}_z$ its Fourier transform. It follows that the electric field component in the $z$ direction is not null and can be written, in Fourier space, as[19]:

$$\tilde{E}^{\omega}_z = -k_z^{-1}(\alpha k_x + j\beta k_y)\tilde{E}_0 \tag{2}$$

where $k_x$ and $k_y$ are the spatial frequencies in the $x$ and $y$ directions, respectively, and $k_z = \sqrt{k_0^2 - k_x^2 - k_y^2}$ is the wavevector component in the $z$ direction, with $k_0$ the free-space wavenumber at frequency $\omega$. In the limit of a weakly divergent pump beam, $k_z \approx k_0$ and with the transformations $jk_x \to \partial/\partial x$ and $jk_y \to \partial/\partial y$, the longitudinal field in real space can be written as follows:

$$E^{\omega}_z \approx \frac{1}{k_0}\left(j\alpha \frac{\partial E_0}{\partial x} - \beta \frac{\partial E_0}{\partial y}\right) = -j\frac{2E_0}{k_0 w^2}(\alpha x + j\beta y) \tag{3}$$

For a circularly-polarized beam, i.e., $\alpha = \sqrt{2}/2$ and $\beta = \pm\sqrt{2}/2$, the longitudinal field shows a donut-shaped amplitude profile, proportional to the radial coordinate $\rho = \sqrt{x^2 + y^2}$. It also acquires a helical phase distribution proportional to the azimuthal coordinate and equal to $\pm\phi = \pm\tan^{-1}(y/x)$, which is a phase singularity at the origin with charge $\pm 1$. This is a well known linear spin-orbit effect that involves only the longitudinal component of

the field[2]. In the linear regime, this interaction has been observed[2] as handedness-dependent transfer of OAM to nanoparticles illuminated by tightly focused, circularly polarized Laguerre-Gaussian beams. The importance of the longitudinal field component extends to nonlinear spin-orbit interactions, as theoretically predicted by Grigoriev et al.[14] for sum-frequency generation within the bulk of isotropic chiral media and isotropic media with nonlinear nonlocal response.

### Spin-orbit coupling involving the longitudinal component in the nonlinear regime

With no loss of generality, let us now explore the role of the longitudinal field in the simple three-wave mixing process of SHG. For AlGaAs, under Kleinmann symmetry, the only nonzero $\chi^{(2)}$ elements are $\chi^{(2)}_{xyz} = \chi^{(2)}_{xzy} = \chi^{(2)}_{yxz} = \chi^{(2)}_{yzx} = \chi^{(2)}_{zxy} = \chi^{(2)}_{zyx}$. Let us consider a thin film of AlGaAs illuminated by a Gaussian pump beam carrying SAM (circularly-polarized) at the fundamental frequency $\omega$. The film is parallel to the ($xy$) plane and located at the waist of the pump beam, where the transverse field can be written as in Eq. (1). To study SHG from the film, we assume that the thickness of the slab is negligible or much smaller than one wavelength ($d \ll \lambda_0$), so that the fields can be approximated as constant within the film and the induced SH source is a nonlinearly polarized sheet at $z = 0$ and with surface susceptibility $\chi^{(2)}_s = \chi^{(2)}d$. The spectral components of the induced polarization on the nonlinear sheet at the SH frequency may be written as:

$$\tilde{P}^{2\omega}_x = \epsilon_0 \chi^{(2)}_s \tilde{E}^{\omega}_y * \tilde{E}^{\omega}_z$$
$$\tilde{P}^{2\omega}_y = \epsilon_0 \chi^{(2)}_s \tilde{E}^{\omega}_x * \tilde{E}^{\omega}_z \tag{4}$$
$$\tilde{P}^{2\omega}_z = \epsilon_0 \chi^{(2)}_s \tilde{E}^{\omega}_x * \tilde{E}^{\omega}_y$$

where $*$ stands for convolution and $\epsilon_0$ is the vacuum permittivity. The transverse components of the SH electric field emitted by the sources in Eq. (4) may be retrieved using the spectral Green function approach outlined in ref. 20 and, when evaluated at $z = 0^{\pm}$, read as follows:

$$\tilde{E}^{2\omega}_x(z = 0^{\pm}) = \tilde{G}_{xx}\tilde{P}^{2\omega}_x + \tilde{G}_{xy}\tilde{P}^{2\omega}_y + \tilde{G}_{xz}\tilde{P}^{2\omega}_z$$
$$\tilde{E}^{2\omega}_y(z = 0^{\pm}) = \tilde{G}_{xy}\tilde{P}^{2\omega}_x + \tilde{G}_{yy}\tilde{P}^{2\omega}_y + \tilde{G}_{yz}\tilde{P}^{2\omega}_z \tag{5}$$

In the limit of a weakly divergent pump beam, $k_z \approx k_0$, $\tilde{G}_{xy} = \tilde{G}_{yx} \approx 0$, $\tilde{G}_{xx} = \tilde{G}_{yy} \approx jk_0/\epsilon_0$, $\tilde{G}_{xz} \approx \mp jk_x/\epsilon_0$, and $\tilde{G}_{yz} \approx \mp jk_y/\epsilon_0$, and therefore the SH fields can be recast as follows:

$$\tilde{E}^{2\omega}_x(z = 0^{\pm}) = jk_0\chi^{(2)}_s \tilde{E}^{\omega}_y * \tilde{E}^{\omega}_z \mp jk_x\chi^{(2)}_s \tilde{E}^{\omega}_x * \tilde{E}^{\omega}_y$$
$$\tilde{E}^{2\omega}_y(z = 0^{\pm}) = jk_0\chi^{(2)}_s \tilde{E}^{\omega}_x * \tilde{E}^{\omega}_z \mp jk_y\chi^{(2)}_s \tilde{E}^{\omega}_x * \tilde{E}^{\omega}_y \tag{6}$$

The expressions of the SH fields in the direct space can be obtained by inverse Fourier transforming Eq. (6):

$$E^{2\omega}_x(z = 0^{\pm}) = jk_0\chi^{(2)}_s E^{\omega}_y E^{\omega}_z \mp \chi^{(2)}_s \frac{\partial(E^{\omega}_x E^{\omega}_y)}{\partial x}$$
$$E^{2\omega}_y(z = 0^{\pm}) = jk_0\chi^{(2)}_s E^{\omega}_x E^{\omega}_z \mp \chi^{(2)}_s \frac{\partial(E^{\omega}_x E^{\omega}_y)}{\partial y} \tag{7}$$

Substitution of Eqs. (3) and (1) in Eq. (7) then yields the following expressions of the transverse SH field:

$$E^{2\omega}_x(z = 0^{\pm}) = j\frac{2\chi^{(2)}_s E_0^2}{w^2}\left[\alpha\beta(1 \pm 2)x + j\,\beta^2 y\right]$$
$$E^{2\omega}_y(z = 0^{\pm}) = j\frac{2\chi^{(2)}_s E_0^2}{w^2}\left[\alpha\beta(1 \pm 2)y - j\,\alpha^2 x\right] \tag{8}$$

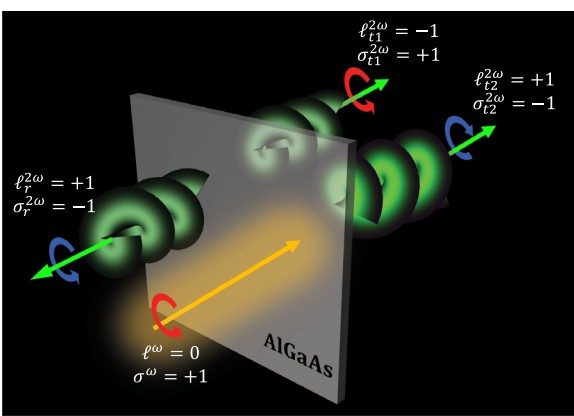

**Fig. 1 | Spin orbit interaction in SHG from a thin film of AlGaAs.** A circularly-polarized pump beam with Gaussian profile illuminates the film at normal incidence. The transmitted SH field is a vortex beam with two components: (i) an LG beam with zero radial number, OAM $\ell_{t1}^{2\omega} = -1$ and SAM $\sigma_{t1}^{2\omega} = +1$, and (ii) an LG beam with 4-fold intensity (the helix reported with the brightest green), zero radial number, OAM $\ell_{t2}^{2\omega} = +1$ and SAM $\sigma_{t2}^{2\omega} = -1$. The reflected SH field is an LG beam with zero radial number, OAM $\ell_r^{2\omega} = +1$ and SAM $\sigma_r^{2\omega} = -1$. The pump beam is represented in orange, while SH field components are in green; red and blue circles indicate left- and right-circular polarization states, respectively.

If we now consider the case of a pump with zero OAM ($\ell^\omega = 0$) and left circular polarization, i.e., $\hat{e} = \hat{L}_+ = (\hat{x} + j\hat{y})/\sqrt{2}$ and therefore SAM equal to $\sigma^\omega = +1$, then the expressions in Eq. (8) reduce to:

$$\mathbf{E}_\perp^{2\omega}(z = 0^+) = j\frac{\sqrt{2}\chi_s^{(2)}E_0^2}{w^2}\rho\left[2e^{j\phi}\hat{R}_+ + e^{-j\phi}\hat{L}_+\right] \qquad (9)$$

at $z = 0^+$, i.e., in transmission, and:

$$\mathbf{E}_\perp^{2\omega}(z = 0^-) = j\frac{\sqrt{2}\chi_s^{(2)}E_0^2}{w^2}\rho\left[-e^{-j\phi}\hat{R}_-\right] \qquad (10)$$

at $z = 0^-$, i.e., in reflection. Here $\rho = \sqrt{x^2 + y^2}$, $\hat{L}_\pm = (\hat{x} \pm j\hat{y})/\sqrt{2}$ indicate the left circular polarization vectors in the forward ($\hat{L}_+$) and backward ($\hat{L}_-$) directions, and $\hat{R}_\pm = (\hat{x} \mp j\hat{y})/\sqrt{2}$ indicate the right circular polarization vectors in the forward ($\hat{R}_+$) and backward ($\hat{R}_-$) directions. The spin orbit transfer described in Eqs. (9) and (10) is illustrated in Fig. 1, in the case of an input Gaussian beam with size $w = 30\,\mu m$ and an input wavelength $\lambda = 1.55\,\mu m$. It is clear that the phase singularity of the longitudinal field at $\omega$ is transferred to the transverse part of the SH field at $2\omega$. Specifically, the reflected SH field is a Laguerre-Gauss (LG) mode carrying SAM with opposite handedness with respect to the exciting pump field, i.e. $\sigma_r^{2\omega} = -1$ (right-circular polarization), and an OAM with charge $\ell_r^{2\omega} = 1$; instead, the transmitted SH field is a superposition of two LG modes, one having SAM with $\sigma_{t1}^{2\omega} = 1$ (left-circular polarization) and OAM with charge $\ell_{t1}^{2\omega} = -1$, and the other one having SAM with $\sigma_{t2}^{2\omega} = -1$ (right-circular polarization) and OAM with charge $\ell_{t2}^{2\omega} = 1$. It is important to notice that the right-circularly polarized component has twice the amplitude of the left-circularly polarized component. Therefore the transmitted SH field has average OAM[21] given by $f_{OAM} = \frac{4\cdot(+1)+1\cdot(-1)}{4+1} = 3/5$ and average SAM $f_{SAM} = \frac{4\cdot(-1)+1\cdot(+1)}{4+1} = -3/5$. The average OAM and SAM values obtained by following the procedure outlined in ref. 21, i.e., by averaging the OAM and SAM with the power carried by each modal component, are identical to those that one finds by using the expressions of the normalized SAM and OAM densities in the longitudinal direction ($z$) based on the canonical momentum density[22-24], which read as follows:

$$J_z^\sigma = \frac{\epsilon_0}{4}\frac{\mathrm{Im}\left[\mathbf{E}^* \times \mathbf{E}\right]}{W} \cdot \hat{z} \qquad (11)$$

$$J_z^\ell = \frac{\epsilon_0}{4}\frac{\mathbf{r} \times \mathrm{Im}\left[\mathbf{E}^* \cdot (\nabla)\mathbf{E}\right]}{W} \cdot \hat{z} \qquad (12)$$

where $W = \frac{1}{4}\epsilon_0|\mathbf{E}|^2$ is the electric energy density, $\mathbf{r}$ is the position vector, and $\mathrm{Im}[\mathbf{E}^* \cdot (\nabla)\mathbf{E}] = \mathrm{Im}[E_x^*\nabla E_x + E_y^*\nabla E_y + E_z^*\nabla E_z]$ represents the canonical momentum of light.

The spatial profiles of transmitted and reflected SH fields predicted by Eqs. (9) and (10) are reported in Fig. 2a, b alongside data of SH generation from an AlGaAs thin film measured with the experimental setup reported in the Methods section (Fig. 2c, d). We find excellent agreement between theory and experiments both in the spatial profile of the field and in the distribution of the polarization state of SH light over the cross section of transmitted and reflected SH vortex beams. The slight discrepancies between the theory and the experiments can be attributed to the effect of the finite thickness of the AlGaAs sample (400 nm), which is not considered in the theoretical predictions. In the calculations, the field is evaluated in the plane of the nonlinear sheet with vanishing thickness. In the experiments, the size and shape of the second harmonic beam slightly vary across the thickness of the sample. Calculated and measured interferograms between the transmitted LG beam and a plane wave confirm the presence of a "fork" dislocation associated with the phase singularity at the center of the beam (see the insets of Fig. 2a, c). We note that the superposition of two LG beams with different intensities in the transmitted SH produces a vortex beam with elliptical polarization mostly in the radial direction. Furthermore, when the pump SAM is switched from $\sigma^\omega = +1$ (left-hand polarization or $\hat{e} = \hat{L}_+$) to $\sigma^\omega = -1$ (right-handed polarization or $\hat{e} = \hat{R}_+$), then the signs of SAM and OAM for SH light flip. Theoretical prediction and experimental verification of this behavior are reported in Fig. 2e, f and Fig 2g, h, respectively.

**Spin orbit coupling with pump beams carrying arbitrary combinations of SAM and OAM**

We now extend the analysis presented in the previous section to the more general case of SHG induced by arbitrary combinations of SAM and OAM in the pump beam. We consider a pump beam with an SAM defined by $\sigma^\omega$ and OAM with charge $\ell^\omega$. The SAM can be $\sigma^\omega = \pm 1$ for circular polarization while the topological charge associated with the OAM state has integer values: $\ell^\omega = \pm 1, \pm 2, \dots$. The expression of the transverse component of the pump electric field at $z = 0$ is then:

$$\mathbf{E}_\perp^\omega(z = 0) = E_0/\sqrt{2} \times LG_0^{\ell^\omega}[w] \times (\hat{x} + j\sigma^\omega\hat{y}) \qquad (13)$$

in which the LG mode has radial index equal to 0 and may be written in the standard way[25]:

$$LG_p^\ell[w] = \left(\frac{\sqrt{2}\rho}{w}\right)^{|\ell|}e^{j\ell\phi}L_p^{|\ell|}\left(\frac{2\rho^2}{w^2}\right) \qquad (14)$$

where $\ell$ is the topological charge, $p$ the radial index and $L_p^{|\ell|}$ the associated Laguerre polynomial. The transverse field of the induced SH in reflection is a single LG mode with amplitude $A_r$, OAM with charge $\ell_r^{2\omega}$, radial index $p_r$, and SAM $\sigma_r^{2\omega}$, and therefore it may be written as follows:

$$\mathbf{E}_\perp^{2\omega}(z = 0^-) = A_r \times LG_{p_r}^{-\ell_r^{2\omega}}[w] \times (\hat{x} - j\sigma_r^{2\omega}\hat{y}) \qquad (15)$$

while the transverse SH field in transmission is a superposition of two LG modes with different amplitudes ($A_{t1}$ and $A_{t2}$), different SAM ($\sigma_{t1}^{2\omega}$ and $\sigma_{t2}^{2\omega}$), different radial index ($p_{t1}$ and $p_{t2}$), and different OAM ($\ell_{t1}^{2\omega}$ and

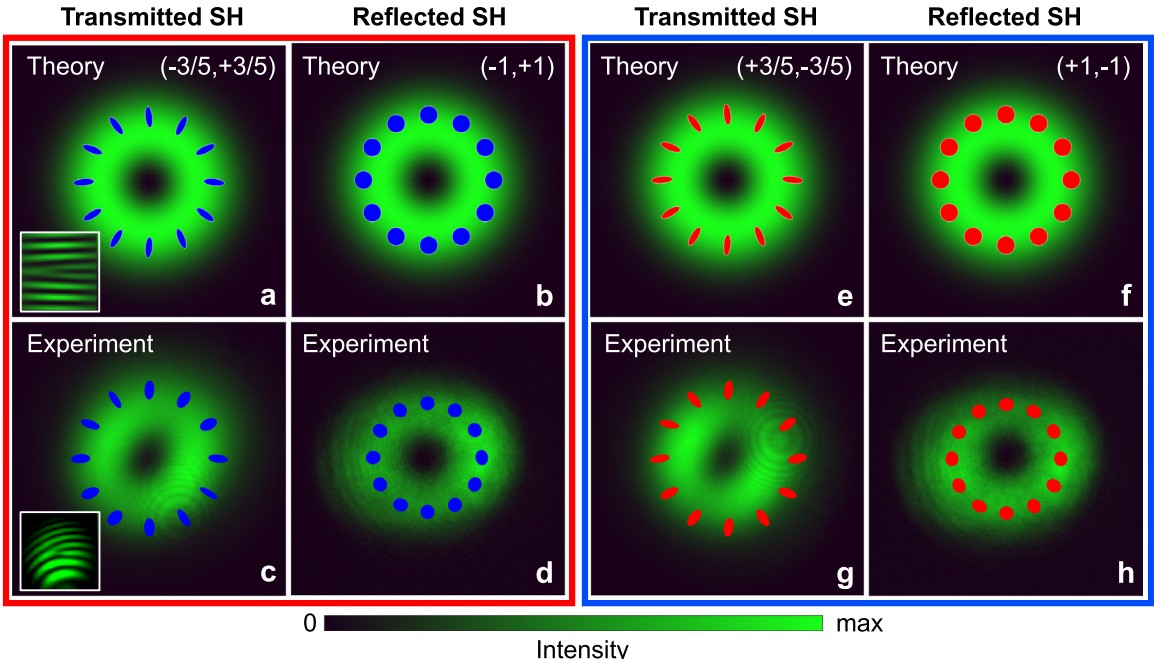

**Fig. 2 | Transmitted and reflected SH beams emitted from an AlGaAs film.**
**a** Intensity distribution ($|\mathbf{E}|^2$) of the transmitted SH vortex beam at $z = 0^+$, theoretically predicted by Eq. (9). The blue ellipses represent the local polarization state across the beam cross section (the blue color indicates right-handed polarization). The inset shows the "fork" dislocation that appears in the plane-wave interferogram associated with the topological charge. **b** Intensity distribution of the reflected SH LG beam at $z = 0^-$, theoretically predicted by Eq. (10). The circles represent the local (circular) polarization state across the beam cross section (the blue color indicates right-handed polarization). **c** Same quantities as those represented in panel **a** but

measured in transmission through an AlGaAs thin film with the experimental setup described in Methods. **d** Same quantities as those represented in panel **b** but measured in reflection from an AlGaAs thin film with the experimental setup described in Methods. In the theoretically predicted plots, the intensity is normalized with respect to the maximum value, and the LG vortex beam is characterized by the pair of angular momentum numbers (SAM,OAM). Plots in the red box refer to the case of SHG with a Gaussian pump carrying SAM given by $\sigma^\omega = +1$, i.e., left-circular polarized. Plots **e**–**h** inside the blue box are the same as **a**–**d**, respectively, for a circularly-polarized Gaussian pump with opposite handedness, i.e., $\sigma^\omega = -1$.

$\ell_{t2}^{2\omega}$), and it has the following expression:

$$
\begin{aligned}
\mathbf{E}_\perp^{2\omega}(z = 0^+) = &A_{t1} \times LG_{p_{t1}}^{\ell_{t1}^{2\omega}}[w] \times (\hat{x} + j\sigma_{t1}^{2\omega}\hat{y}) + \\
&+ A_{t2} \times LG_{p_{t2}}^{\ell_{t2}^{2\omega}}[w] \times (\hat{x} + j\sigma_{t2}^{2\omega}\hat{y})
\end{aligned}
\tag{16}
$$

In Eqs. (15) and (16), the amplitudes of the transmitted and reflected LG modes are:

$$
A_{t1} = j\frac{\chi_S^{(2)}}{w}E_0^2 s_{t1}
\tag{17}
$$

$$
A_{t2} = 2j\frac{\chi_S^{(2)}}{w}E_0^2 s_{t2}
\tag{18}
$$

$$
A_r = j\frac{\chi_S^{(2)}}{w}E_0^2 s_r
\tag{19}
$$

where $s_r = s_{t2} = -s_{t1} = \text{sgn}(\ell^\omega)$, while the SAM and OAM of transmitted (the two LG modes indicated with subscripts $t1$ and $t2$) and reflected (the LG mode indicated with subscript $r$) SH beams are given by the following rules:

$$
\sigma_{t1}^{2\omega} = \sigma^\omega \qquad \ell_{t1}^{2\omega} = 2\ell^\omega - \sigma^\omega
\tag{20}
$$

$$
\sigma_{t2}^{2\omega} = -\sigma^\omega \qquad \ell_{t2}^{2\omega} = 2\ell^\omega + \sigma^\omega
\tag{21}
$$

$$
\sigma_r^{2\omega} = -\sigma^\omega \qquad \ell_r^{2\omega} = -(2\ell^\omega - \sigma^\omega)
\tag{22}
$$

The radial index of LG modes at the second-harmonic can be either 0 (conservation of the radial index of the pump) or 1 (radial index increased with respect to the pump). In particular, the radial index is $p_{r,t1,t2} = 1$ for modes in which $|\ell_{r,t1,t2}^{2\omega}| = 2|\ell^\omega| - 1$, and $p_{r,t1,t2} = 0$ for modes in which $|\ell_{r,t1,t2}^{2\omega}| = 2|\ell^\omega| + 1$. The appearance of a higher radial order has been observed in type II second-harmonic generation with orbital-angular-momentum mixing of counter-rotating vortices[26], in spontaneous-parametric down conversion[27] and in self-phase modulation[28], and it has been interpreted as a diffraction effect in[29]. The peculiarity of nonlinear spin-orbit coupling in AlGaAs is that the appearance of a higher radial order is strictly related to the presence of spin in the pump beam, as well as to the $\chi^{(2)}$ tensor symmetry.

From Eqs. (20) and (21), it turns out that the average SAM and average OAM[21] of the vortex beam generated in transmission are:

$$
f_{\text{SAM}} = \frac{\sigma_{t1}^{2\omega}|A_{t1}|^2 + \sigma_{t2}^{2\omega}|A_{t2}|^2}{|A_{t1}|^2 + |A_{t2}|^2} = -\frac{3}{5}\sigma^\omega
\tag{23}
$$

$$
f_{\text{OAM}} = \frac{\ell_{t1}^{2\omega}|A_{t1}|^2 + \ell_{t2}^{2\omega}|A_{t2}|^2}{|A_{t1}|^2 + |A_{t2}|^2} = 2\ell^\omega + \frac{3}{5}\sigma^\omega
\tag{24}
$$

To verify these predictions, second-harmonic generation on the transmission side has been measured for all possible combinations of SAM and OAM states of the pump achievable with $\ell^\omega = 0, \pm 1$ and $\sigma^\omega = 0, \pm 1$. We opted to image the second harmonic beams in the Fourier space, by inserting a Bertrand lens in the setup (see the Experimental setup section). This configuration is equivalent to visualizing the beams far away from the sample surface, in the Fraunhofer approximation. At such a distance, the differences

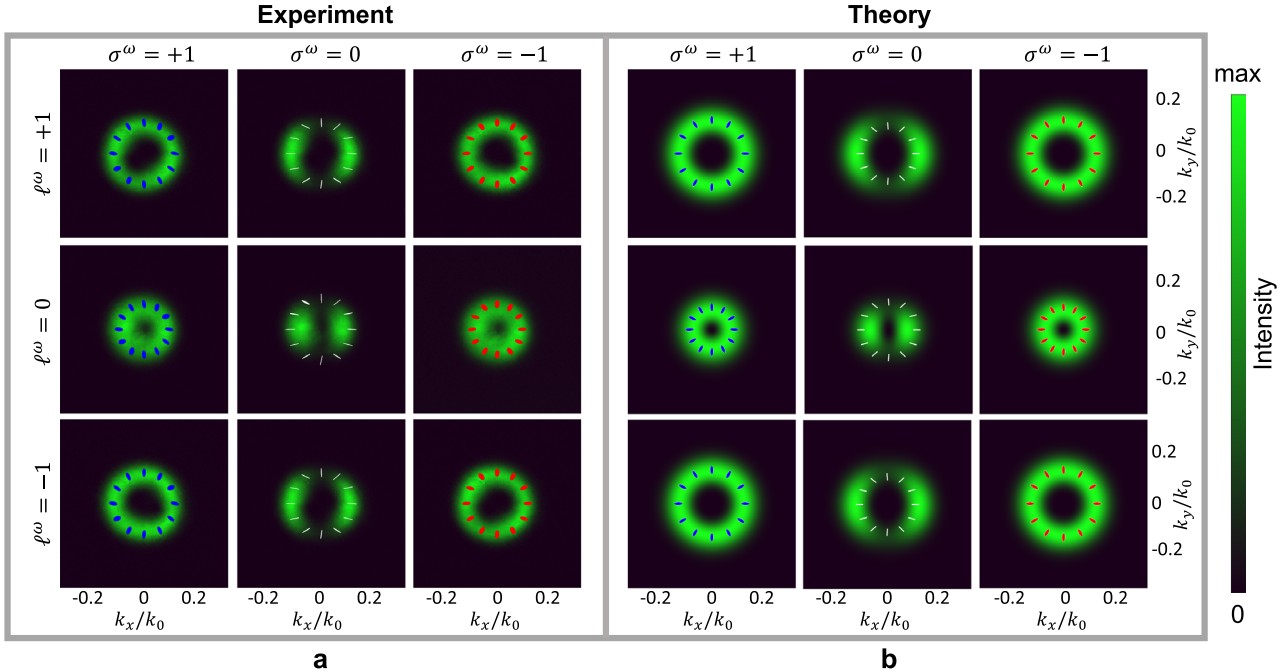

**Fig. 3 | Nonlinear spin orbit coupling with pump beams carrying arbitrary combinations of SAM and OAM.** Experimental (**a**) and theoretical (**b**) spatial profiles of the SH beam intensity for the nine combinations of pump SAM and OAM states achievable with $\ell^{\omega} = 0, \pm 1$ and $\sigma^{\omega} = 0, \pm 1$. The profiles are measured (**a**) and calculated (**b**) in the Fourier plane on the transmission side, as described in Methods (Experimental setup section). The distribution of the polarization state across the beam is illustrated with red (left-handed polarization) and blue (right-handed polarization) ellipses. Linear polarization states that occur for $\sigma^{\omega} = 0$ are indicated with white segments.

between the infinitely thin sheet of nonlinear material considered in the theory and the experimental sample, with a small but finite thickness, can be neglected. The results are presented in Fig. 3. The experimental findings, including the spatial distribution of the measured second-harmonic intensity and the polarization state across the beam in the Fourier plane, are juxtaposed with corresponding theoretical predictions, showing excellent agreement. A larger subset of the possible nonlinear spin orbit interactions in AlGaAs is reported in Table 1 for a few combinations of SAM and OAM states of the pump.

For illustrative purposes, the pump, as well as the SH beams generated in transmission and reflection, calculated in the direct space according to Eq. (16), are reported in Fig. 4 for all the interactions reported in Table 1.

**Table 1 | Nonlinear spin orbit interactions in AlGaAs (001)**

| Pump | | SH$_{t1}$ | | SH$_{t2}$ | | SH$_{t1}$+ SH$_{t2}$ | | SH$_r$ | |
|---|---|---|---|---|---|---|---|---|---|
| $\sigma^{\omega}$ | $\ell^{\omega}$ | $\sigma_{t1}^{2\omega}$ | $\ell_{t1}^{2\omega}$ | $\sigma_{t2}^{2\omega}$ | $\ell_{t2}^{2\omega}$ | $f_{SAM}$ | $f_{OAM}$ | $\sigma_r^{2\omega}$ | $\ell_r^{2\omega}$ |
| 0 | 1 | 0 | 2 | 0 | 2 | 0 | 2 | 0 | −2 |
| 1 | 0 | 1 | −1 | −1 | 1 | −3/5 | 3/5 | −1 | 1 |
| 1 | −1 | 1 | −3 | −1 | −1 | −3/5 | −7/5 | −1 | 3 |
| 1 | 1 | 1 | 1 | −1 | 3 | −3/5 | 13/5 | −1 | −1 |
| 1 | −2 | 1 | −5 | −1 | −3 | −3/5 | −17/5 | −1 | 5 |
| 1 | 2 | 1 | 3 | −1 | 5 | −3/5 | 23/5 | −1 | −5 |

Nonlinear spin orbit interactions in AlGaAs for selected combinations of SAM and OAM states of the pump. The first and second column refer to the SAM and OAM of the pump (labeled "Pump"); the second and third columns refer to the SAM and OAM of the first SH component in transmission (labeled "SH$_{t1}$"), whose relative amplitude is equal to 1; the fifth and sixth columns refer to the SAM and OAM of the second SH component in transmission (labeled "SH$_{t2}$"), whose relative amplitude is equal to 2; the seventh and eighth columns are the average SAM and average OAM of the overall SH vortex beam generated in transmission; the last two columns refer to the SAM and OAM of the reflected SH beam (labeled "SH$_r$"), whose relative amplitude is −1.

## Discussion

We have presented theoretical and experimental evidence of a new type of spin-orbit coupling occurring in second-harmonic generation from thin films. The nonlinear interaction is mediated by the longitudinal component of the pump field, and therefore it requires materials with nonlinear susceptibility tensor elements that mix longitudinal and transverse components of the pump. We have demonstrated that AlGaAs thin films, when pumped with circularly-polarized light, emit vortex beams of second-harmonic light carrying both spin- and orbital angular momentum. In particular, average SAM and OAM are produced in transmission due to the superposition of LG beams with opposite topological charge and different intensities. We have provided general rules to determine SAM and OAM of SH light in the presence of different combinations of SAM and OAM pump states. These rules are related to the particular choice of the shape of the second-order nonlinear tensor, and therefore we foresee that this type of spin orbit coupling may be engineered in structured films, i.e., metasurfaces, to produce a variety of OAM states. Besides improving our understanding of fundamental aspects of vectorial nonlinear optics, our results suggest that the nonlinear response of seemingly simple systems such as non-structured thin films can be leveraged to structure nonlinearly scattered light and may find applications in a variety of fields, such as quantum optics, information processing and communications, optical trapping and imaging.

## Methods

### Fabrication of the AlGaAs thin-film sample

Sample fabrication begins with the growth of a planar structure by molecular beam epitaxy on a GaAs (001) wafer. The structure consists of a 400 nm thick layer of $Al_{0.17}Ga_{0.83}As$ which rests on a sacrificial layer of $Al_{0.8}Ga_{0.2}As$. $Al_{0.17}Ga_{0.83}As$ is transparent at the SH frequency and, at the same time, is immune to two-photon absorption at the pump wavelength. Cleaved parts of the sample are then glued upside

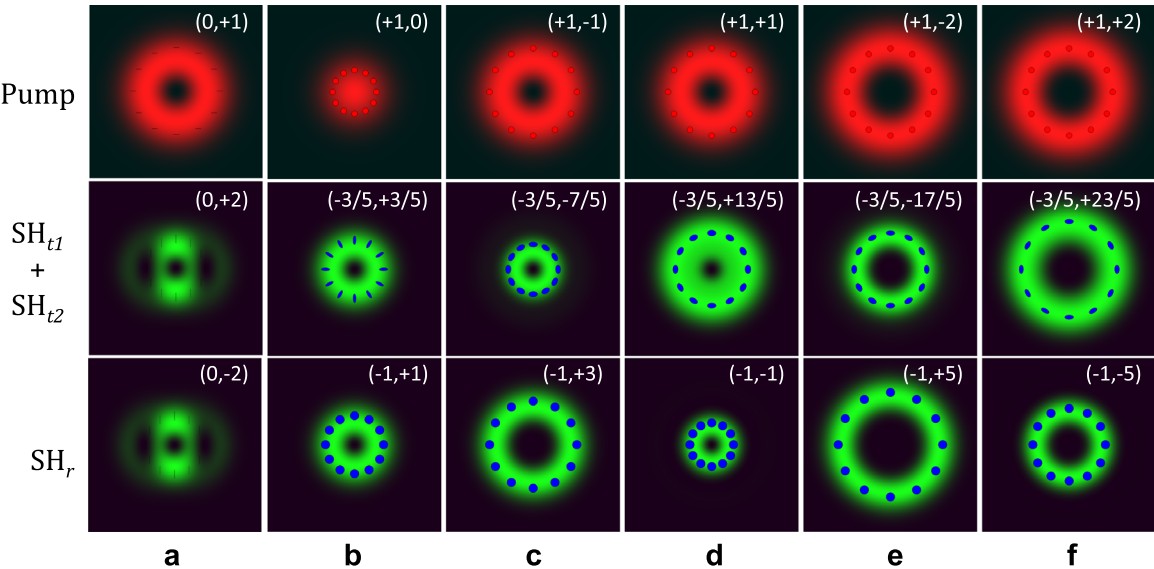

**Fig. 4 | Cross sections of pump and SH beams for selected combinations of SAM and OAM states of the pump.** Panels **a**–**f** illustrate the cross-sections of the pump (first row), transmitted (second row) and reflected (third row) SH fields associated with all the interactions described in the six rows of Table 1. All the cross sections are calculated at $z = 0^{\pm}$ and in each plot, the LG vortex beam is characterized by the pair of angular momentum numbers (SAM,OAM), as reported in Table 1. **a** The pump state is (SAM,OAM) = (0,+1). **b** The pump state is (SAM,OAM) = (+1,0). **c** The pump state is (SAM,OAM) = (+1,−1). **d** The pump state is (SAM,OAM) = (+1,+1). **e** The pump state is (SAM,OAM) = (+1,−2). **f** The pump state is (SAM,OAM) = (+1,−2).

down to a sapphire substrate using an epoxy adhesive (M-bond 610). Subsequently, a series of mechanical etchings using abrasive discs with different roughness levels is employed to partially remove the GaAs (001) substrate which was utilized during MBE growth. The polishing is stopped when its thickness reaches approximately 35 μm, so as to avoid the formation of structural defects in the active layer. We take off the residual GaAs layer through a selective chemical etching (citric acid and $H_2O_2$ diluted in deionized water) with a typical GaAs etch rate of ≈1.5 μm/min. As a final step, we remove the sacrificial $Al_{0.8}Ga_{0.2}As$ layer with a second selective chemical etch using concentrated hydrofluoric acid. During this final step, only the sacrificial layer undergoes etching, thus leaving behind a flat, clean and mirror-like $Al_{0.17}Ga_{0.83}As$ membrane glued on the transparent sapphire substrate.

### Experimental setup

The nonlinear characterization setup is based on a horizontal microscope presented in Supplementary Fig. 1. The pump field is generated by an optical parametric amplifier (Mango, APE) pumped by a mode-locked Yb-doped fiber laser (Satsuma, Amplitude). This source emits 160 fs-pulses at 1550 nm with a repetition rate of 500 kHz. The pump beam size is controlled by a telescope, its polarization state is set by a linear polarizer and a quarter-wave plate and the pump OAM is tuned by reflecting the beam on an SLM (200-21, Santec). The pump power is controlled with a half-wave plate and a Glan-Taylor polarizer, and then it is measured by sending a fraction of it on an InGaAs photodiode through beam-splitter BS2. The pump beam is focused on the sample by a micro-lens with NA = 0.16 and high transmission at the fundamental frequency. The reflected (resp. transmitted) pump beam is removed by a short-pass dichroic mirror (resp. short-pass filter). After being collected by a micro-lens (NA = 0.68), the forward "transmitted" SH signal is focused either on a CCD camera (Sony, ICX825AL), or on a spectrometer (Broadcom, Qmini), or on a Si photodiode (Newport, 818-SL). For SHG measurements in transmission, we move from real to Fourier space by imaging the back focal plane of the collection lens with a Bertrand lens. For phase imaging of the SH signal, we let it interfere with a reference beam at SH in a Mach-Zender interferometer that includes beam splitters BS1 and BS3. The former sends a part of the pump beam to the reference path, where first a phase-matched BBO crystal doubles its frequency; then a delay line, consisting of a micrometric precision motorized stage, ensures the temporal overlap of the two pulses on BS3. The resulting interference pattern is recorded on the CCD, while the transmitted reference and reflected sample beams at BS3 are sent to a beam dump.

### SH characterization

Spectral properties of the SH signal were characterized using a fiber coupled spectrometer. Results are presented in Supplementary Fig. 2, where the Gaussian fit gives a peak wavelength of 775.2 nm and a full width at half maximum of 5.5 nm.

### SHG efficiency measurement

SH power scales with the square of the pump power. Thus, for low incident power, we implemented lock-in detection to measure low SH power. A mechanical chopper modulates the pump intensity and serves as reference input for the lock-in amplifier. We simultaneously measure the ratio $\overline{P}_{SH}/\overline{P}_{FF}$ and the results are presented in Supplementary Fig. 3. We report maximum efficiency of $8.2 \times 10^{-8}$ for an average pump power of 16.8 mW. As expected for second order nonlinear process, the fitting coefficient is $2.02 \pm 0.04$.

### Stokes parameters measurement

The polarization of a beam with intensity $I$ and degree of polarization $p$ can be characterized in terms of four intensity parameters:

$$S = \begin{pmatrix} I \\ Q \\ U \\ V \end{pmatrix} = \begin{pmatrix} I \\ Ip\cos(2\psi)\cos(2\chi) \\ Ip\sin(2\psi)\cos(2\chi) \\ Ip\sin(2\chi) \end{pmatrix} \tag{25}$$

where $\psi$, the inclination, gives the orientation of the polarization ellipse and $\chi$, the ellipticity, gives the ratio between the two semi-axes.

The measurement of such parameters is carried out by adding two polarizing elements in the optical path just before the imaging camera. As discussed in ref. 30, we use a quarter-waveplate that can be rotated by an angle $\theta$ followed by a fixed wire grid polarizer whose

transmission axis is aligned with the x-axis. These two elements are placed in the transmission arm of the setup (QWP2 and LP2) or in the reflection arm (QWP3 and LP3).

The intensity is given by

$$I(\theta) = \frac{1}{2}(A + B\sin 2\theta + C\cos 4\theta + D\sin 4\theta) \qquad (26)$$

Experimentally, we acquire an intensity map of the SH beam for $N$ discrete values of $\theta$. Eq. (26) needs to be rewritten as

$$I_n = \frac{1}{2}(A + B\sin 2\theta_n + C\cos 4\theta_n + D\sin 4\theta_n) \, (n = 1, 2, \cdots, N)(N \geq 8) \qquad (27)$$

where $N$ is an even number whose minimum value is given by Nyquist's sampling theorem. Thus, for eight measurements with equal angle intervals, we can recover A,B,C and D and the Stokes parameters

$$A = \frac{2}{N}\sum_{n=1}^{N} I_n, \qquad B = \frac{4}{N}\sum_{n=1}^{N} I_n \sin 2\theta_n,$$
$$C = \frac{4}{N}\sum_{n=1}^{N} I_n \cos 4\theta_n, \qquad D = \frac{4}{N}\sum_{n=1}^{N} I_n \sin 4\theta_n \qquad (28)$$

$$I = A - C, Q = 2C, U = 2D, V = B \qquad (29)$$

$$p = \frac{1}{I}\sqrt{Q^2 + U^2 + V^2}$$
$$\text{and, for } p = 1:$$
$$\psi = \frac{1}{2}\tan^{-1}\left(\frac{U}{Q}\right) \qquad (30)$$
$$\chi = \frac{1}{2}\tan^{-1}\left(\frac{V}{I}\right)$$

As an example, we show in Supplementary Fig. 4 the experimental distribution of $p, \chi$ and $\psi$ characterizing the polarization state of the SH beams of Fig. 2c, d. The blue ellipses in Fig. 2c, d are derived from these experimental maps.

## Data availability
The data that support the findings of this study are available from the corresponding authors upon request.

## Code availability
The codes that support the figures of this paper are available from the corresponding authors upon request.

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

## Acknowledgements
This work was supported by "RESPONDER" Project NATO SPS G5984 (M.A.V and D.d.C.), "METAFAST" H2020-FETOPEN-2018-2020 grant agreement no. 899673 (C.D.A.), Ministero Italiano dell'Istruzione (MIUR)

through the "METEOR" project PRIN-2020 2020EY2LJT_002 (C.D.A.), French Agence Nationale de la Recherche through the NOMOS (ANR-18-CE24-0026) and IPOD (ANR-19-CE47-0009) projects grants (G.L., J.M.G., J.C. and R.T.), ERC "FORWARD" grant agreement no. 771688 (A.D. and I.R.); French Agence de l'Innovation de Défense (50% of L.C.'s PhD grant). The authors thank Yann Genuist (CNRS/Institut Néel) for the growth of the AlGaAs layers by molecular beam epitaxy.

## Author contributions

D.d.C. and C.D.A. conceived the idea. D.d.C., M.A.V. and M.S. performed the numerical simulations. R.T., J.C. and J.M.G. fabricated the device. L.C., I.R., A.D. and G.L designed the experimental setup. L.C. performed the measurements under the supervision of A.D. and G.L. All authors analyzed the data. D.d.C and L.C. wrote the manuscript, with inputs from all authors.

## Competing interests

The authors declare no competing interest.
