## [Peer Review File · Nature Communications]

Nonlinear spin-orbit coupling in optical thin filmsReviewer #1 (Remarks to the Author):

Spin-orbit optical interactions, involving the coupling between the polarization (spin) and the spatial phase distribution (orbital) of an optical wave, have garnered significant attention in recent years due to their foundational implications and potential applications in fields such as microscopy, optical micro-manipulation, and quantum information (see, for example, the review referenced as [1] in the manuscript).

This paper presents what is, to the best of my knowledge, the first observation of a new mechanism for light's spin-orbit interaction within a medium. Initially, this spin-orbit coupling occurs in the input field due to well-established linear-optical mechanisms involving non-paraxiality. Specifically, the spin angular momentum of the incoming wave is converted into the orbital angular momentum (OAM) of the longitudinal field component, a conversion most evident near the beam's focus. However, when operating in isolation, this mechanism cannot produce far-field effects. As one moves to the far-field, the longitudinal field component diminishes, and the role of spin-orbit coupling fades. In this work, the medium optical nonlinearity serves as the coupling agent between the longitudinal component of the input field and the transverse components of the radiated second-harmonic waves. This coupling hence enables the OAM to be transferred to the transverse field components of the second harmonic and hence propagate beyond the focal region into the far-field. To achieve this, the medium must have a nonzero $\chi^{(2)}_{xyz}$ tensor element (and possible index permutations), which is allowed only under specific medium symmetry properties that are relatively uncommon.

A similar theoretical concept for spin-orbit coupling was previously outlined in Ref. [14] of the manuscript, considering a chiral isotropic material as optical medium. Chirality, however, was not essential to the phenomenon described in Ref. [14]; it merely provided the symmetry conditions for having a nonzero $\chi^{(2)}_{xyz}$ tensor element. To my knowledge, the reported phenomenon had not been experimentally observed in any medium prior to this work. In the current manuscript, the authors have selected gallium arsenide—a medium with a nonzero $\chi^{(2)}_{xyz}$ term independent of chirality—as the material of study. This choice has enabled them to provide robust experimental verification of the phenomenon, backed by a comprehensive set of experiments. Similar to other spin-orbit effects, the reported phenomenon enables the generation of vortex waves from a circularly polarized Gaussian beam of light. The paper is well-crafted, and the presented work appears to be sound.

In conclusion, this paper offers the first observation of an intriguing new mechanism within the realm of spin-orbit optical interactions. I believe this result merits publication in Nature Communications. However, before proceeding to publication, I recommend that the authors more clearly acknowledge the relevance of the theoretical results of Ref. [14] for their work.

Reviewer #2 (Remarks to the Author):

Report on manuscript: Nonlinear spin-orbit coupling in optical thin films

The authors report the observation of spin to orbital angular momentum transfer in optical second harmonic generation produced by an AlGaAs thin film, which converts a 1550nm pump photons into 775nm second harmonic photons. The spin-orbit coupling occurs through the contribution of the longitudinal electric field component of the pump beam to the nonlinear response of the film. This longitudinal component is theoretically calculated by imposing the zero divergence of the electric field. All three components of the electric polarization in the film are then obtained from the nonlinear susceptibility tensor. In this way, the authors predict the appearance of three second harmonic beams, two transmitted and one reflected by the film surface. When the pump beam is prepared with a Gaussian mode ($l_w = 0$) and left circular polarization ($\sigma_w = +1$), one transmitted beam carries topological charge $l_{2w} = -1$ and left circular polarization ($\sigma_{2w} = +1$), the other carries $l_{2w} = +1$ and right circular polarization ($\sigma_{2w} = -1$), and the reflected beam carries $l_{2w} = +1$ and right circular polarization ($\sigma_{2w} = -1$). The corresponding experimental results agree very well with the theoretical predictions.

Then, the authors generalize their theory to arbitrary combinations of spin and orbital angular momenta in the pump beam, leading to the conditions given in Eqs. (20)-(22) and Table I. Their theory also predicts the relative amplitudes of the three SH beams generated in the process, from which they calculate the fractional values of the average spin and orbital angular momenta. No experimental results are provided for this general case.

In my opinion, the theoretical and experimental results shown for the cases with $\sigma_w = \pm 1$ and $l_w = 0$ are interesting and certainly deserve publication. However, the the lack of experimental results for the cases with $l_w \neq 0$ weakens the claim for publication in Nature Communications. Moreover, there are some issues regarding the presentation and the physical interpretation of their results, which need to be carefully addressed before a final decision is made:

1- The description of the experimental setup is quite confusing. First, the authors mention a Bertrand lens (BL) in the caption of Fig. 4, but neither it is identified in the figure, nor its role discussed in the text. Second, the BS's should be numbered and their orientations carefully revised. I have the impression that the second BS on the 1550nm beam is wrong. It is also confusing what happens in the BS placed in the SH beam, which splits it at the bottom of the figure. It is not clear why two polarization measurement setups (QWP2-LP2 and QWP3-LP3) are needed.

2- The results presented in the manuscript clearly demonstrate that the angular momentum is not conserved in the process, which is natural due to the anisotropy of the nonlinear medium. The authors should comment on that and make a quick estimate of the torque exerted on the film. As a thin layer, maybe it is not completely impossible to devise a measurement scheme in a future work.

3- The authors should also consider the use of vector beams (radially or azimuthally polarized) since their longitudinal electric field component is increased as compared to the usual scalar beams.

4- I do not agree with the terminology used for the quantities calculated in Eqs. (23) and (24). There is a literature about fractional angular momentum, which is related to different kinds of phase and polarization structures. For example, the authors should look at PRL 95, 240501 (2005) and Sci. Adv. 9, eadf3486 (2023). The quantity calculated by the authors is more related to the average spin and angular momenta since they come from the weighted sum of these quantities in the transmitted light.

Manuscript: NCOMMS-23-43699

Title: Nonlinear spin-orbit coupling in optical thin-films

Authors: Domenico de Ceglia, Laure Coudrat, Iannis Roland, Maria Antonietta Vincenti, Michael Scalora, Rana Tanos, Julien Claudon, Jean-Michel Gerard, Aloyse Degiron, Giuseppe Leo, and Costantino De Angelis

Response to Reviewers

Reviewer 1: *Spin-orbit optical interactions, involving the coupling between the polarization (spin) and the spatial phase distribution (orbital) of an optical wave, have garnered significant attention in recent years due to their foundational implications and potential applications in fields such as microscopy, optical micro-manipulation, and quantum information (see, for example, the review referenced as [1] in the manuscript).*

This paper presents what is, to the best of my knowledge, the first observation of a new mechanism for light's spin-orbit interaction within a medium. Initially, this spin-orbit coupling occurs in the input field due to well-established linear-optical mechanisms involving non-paraxiality. Specifically, the spin angular momentum of the incoming wave is converted into the orbital angular momentum (OAM) of the longitudinal field component, a conversion most evident near the beam's focus. However, when operating in isolation, this mechanism cannot produce far-field effects. As one moves to the far-field, the longitudinal field component diminishes, and the role of spin-orbit coupling fades. In this work, the medium optical nonlinearity serves as the coupling agent between the longitudinal component of the input field and the transverse components of the radiated second-harmonic waves. This coupling hence enables the OAM to be transferred to the transverse field components of the second harmonic and hence propagate beyond the focal region into the far-field. To achieve this, the medium must have a nonzero $\chi^{(2)}$ tensor element (and possible index permutations), which is allowed only under specific medium symmetry properties that are relatively uncommon.

Authors' reply: We thank the Reviewer for carefully reading our manuscript and for perfectly describing the mechanism of nonlinear spin-orbit coupling in the system that we have investigated.

Reviewer 1: *A similar theoretical concept for spin-orbit coupling was previously outlined in Ref. [14] of the manuscript, considering a chiral isotropic material as optical medium. Chirality, however,*

was not essential to the phenomenon described in Ref. [14]; it merely provided the symmetry conditions for having a nonzero $\chi^{(2)}_{xyz}$ tensor element. To my knowledge, the reported phenomenon had not been experimentally observed in any medium prior to this work. In the current manuscript, the authors have selected gallium arsenide—a medium with a nonzero $\chi^{(2)}_{xyz}$ term independent of chirality—as the material of study. This choice has enabled them to provide robust experimental verification of the phenomenon, backed by a comprehensive set of experiments. Similar to other spin-orbit effects, the reported phenomenon enables the generation of vortex waves from a circularly polarized Gaussian beam of light. The paper is well-crafted, and the presented work appears to be sound.

In conclusion, this paper offers the first observation of an intriguing new mechanism within the realm of spin-orbit optical interactions. I believe this result merits publication in Nature Communications. However, before proceeding to publication, I recommend that the authors more clearly acknowledge the relevance of the theoretical results of Ref. [14] for their work.

Authors' reply: We thank the Reviewer for recognizing the novelty in our work. We also agree with the Reviewer that in Ref. 14 the role of the longitudinal component in nonlinear spin orbit coupling has been theoretically investigated in different material systems, namely isotropic chiral media and isotropic media with nonlocal nonlinear response, and that there is some connection with some aspects of our work. To make this clearer for the reader, we have included this sentence at the end of sec. 2.1:

“The importance of the longitudinal field component extends to nonlinear spin-orbit interactions, as theoretically predicted by Grigoriev et al. in [14] for sum-frequency generation within the bulk of isotropic chiral media and isotropic media with nonlinear nonlocal response.”

Reviewer 2: *The authors report the observation of spin to orbital angular momentum transfer in optical second harmonic generation produced by an AlGaAs thin film, which converts a 1550nm pump photons into 775nm second harmonic photons. The spin-orbit coupling occurs through the contribution of the longitudinal electric field component of the pump beam to the nonlinear response of the film. This longitudinal component is theoretically calculated by imposing the zero divergence of the electric field. All three components of the electric polarization in the film are then obtained from the nonlinear susceptibility tensor. In this way, the authors predict the appearance of three*

second harmonic beams, two transmitted and one reflected by the film surface. When the pump beam is prepared with a Gaussian mode ($l_w = 0$) and left circular polarization ($\sigma_w = +1$), one transmitted beam carries topological charge $l_{2w} = -1$ and left circular polarization ($\sigma_{2w} = +1$), the other carries $l_{2w} = +1$ and right circular polarization ($\sigma_{2w} = -1$), and the reflected beam carries $l_{2w} = +1$ and right circular polarization ($\sigma_{2w} = -1$). The corresponding experimental results agree very well with the theoretical predictions.

Then, the authors generalize their theory to arbitrary combinations of spin and orbital angular momenta in the pump beam, leading to the conditions given in Eqs. (20)-(22) and Table I. Their theory also predicts the relative amplitudes of the three SH beams generated in the process, from which they calculate the fractional values of the average spin and orbital angular momenta. No experimental results are provided for this general case.

In my opinion, the theoretical and experimental results shown for the cases with $\sigma_w = \pm 1$ and $l_w = 0$ are interesting and certainly deserve publication. However, the the lack of experimental results for the cases with $l_w \neq 0$ weakens the claim for publication in Nature Communications.

Authors' reply: We thank the Reviewer for the positive assessment of the results reported in our manuscript and for stating that our results are interesting and deserve publication.

We also thank the Reviewer for suggesting to improve our manuscript by performing additional experiments with $\ell^\omega \neq 0$. Motivated by the Reviewer's advice, we have equipped our setup with a spatial light modulator and analyzed cases in which the pump carries both OAM and SAM. The results of these new experiments are in excellent agreement with the theoretical predictions. A side-by-side comparison of measured and theoretically predicted SH generation is presented in Fig. 3 of the revised manuscript. In this new figure, we show both the spatial profile and the polarization state across the SH beam cross section for the nine combinations of OAM/SAM pump states that are achievable with $\ell^\omega = 0, \pm 1$ and $\sigma^\omega = 0, \pm 1$. For this set of measurements, we opted to image the second harmonic beams in the Fourier space, by inserting a Bertrand lens in the setup (see section 4.2). This configuration is equivalent to visualizing the beams far away from the sample surface, in the Fraunhofer approximation. At such a distance, the differences between the infinitely thin sheet of nonlinear material considered in the theory and the experimental sample, with a small but finite thickness, can be neglected. Indeed, the slight discrepancies between theoretically predicted and

measured SH vortex beams that are observable in direct space (see Fig. 2), virtually vanish in Fourier space (see Fig. 3).

The results in Fig. 3 are discussed in section 2.3, where the following text has been added in the revised manuscript:

“To verify these predictions, second-harmonic generation on the transmission side has been measured for all possible combinations of SAM and OAM states of the pump achievable with $\ell^\omega = 0, \pm 1$ and $\sigma^\omega = 0, \pm 1$. We opted to image the second harmonic beams in the Fourier space, by inserting a Bertrand lens in the setup (see section 4.2). This configuration is equivalent to visualizing the beams far away from the sample surface, in the Fraunhofer approximation. At such a distance, the differences between the infinitely thin sheet of nonlinear material considered in the theory and the experimental sample, with a small but finite thickness, can be neglected. The results are presented in Fig. 3. The experimental findings, including the spatial distribution of the measured second-harmonic intensity and the polarization state across the beam in the Fourier plane, are juxtaposed with corresponding theoretical predictions, showing excellent agreement.”

In section 2.2, we have included the following sentences to better describe the results in Fig. 2: “The slight discrepancies between the theory and the experiments can be attributed to the effect of the finite thickness of the AlGaAs sample (400 nm), which is not considered in the theoretical predictions. In the calculations, the field is evaluated in the plane of the nonlinear sheet with vanishing thickness. In the experiments, the size and shape of the second harmonic beam slightly vary across the thickness of the sample.”

Next we have improved our analysis of spin-orbit interactions in the case of an input pump carrying nonzero spin and orbital angular momentum and zero radial index. In particular, we have found that a change of radial index from 0 to 1 is possible for LG second-harmonic beams emitted with an OAM that satisfies the condition $|\ell^{2\omega}| = 2|\ell^\omega| - 1$. While this phenomenon does not modify the computation of spin and angular momentum in reflection and transmission, it introduces a significant aspect that underscores the richness of physical effects observable within this seemingly simple system. This addition contributes to a deeper understanding of the complexity inherent in nonlinear spin-orbit interactions and enriches our comprehension of the diverse phenomena they can exhibit.

The following modifications have been implemented in section 2.2 to address this point (all modifications are marked in red in the revised manuscript):

- Eqs. 14-16 have been updated by adding the radial index in the LG term, which in general can be different from 0, and after Eq. 14 we added the following sentence: “where ℓ is the topological charge and p the radial index”

- Sign prefactors in front of the expressions of the amplitudes of the LG second-harmonic modes in Eqs. 17-19 have been introduced s_r, s_{t1}, s_{t2} , which are instrumental to determine the initial phase of the LG modes even for nonzero radial index. The values of these prefactors are provided after Eq. 19, i.e., $s_r = s_{t2} = -s_{t1} = \text{sgn}(\ell^\omega)$.

- We added the following sentence on page 9: “The radial index of LG modes at the second-harmonic can be either 0 (conservation of the radial index of the pump) or 1 (radial index increased with respect to the pump). In particular, the radial index is $p_{r,t1,t2} = 1$ for modes in which $|\ell_{r,t1,t2}^{2\omega}| = 2|\ell_{r,t1,t2}^\omega| - 1$, and $p_{r,t1,t2} = 0$ for modes in which $|\ell_{r,t1,t2}^{2\omega}| = 2|\ell_{r,t1,t2}^\omega| + 1$ ”

Reviewer 2: *Moreover, there are some issues regarding the presentation and the physical interpretation of their results, which need to be carefully addressed before a final decision is made:*

1- The description of the experimental setup is quite confusing. First, the authors mention a Bertrand lens (BL) in the caption of Fig. 4, but neither it is identified in the figure, nor its role discussed in the text. Second, the BS's should be numbered and their orientations carefully revised. I have the impression that the second BS on the 1550nm beam is wrong. It is also confusing what happens in the BS placed in the SH beam, which splits it at the bottom of the figure. It is not clear why two polarization measurement setups (QWP2-LP2 and QWP3-LP3) are needed.

Authors' reply: We thank the reviewer for his comments on the experimental methods. We have revised Fig. 4 and its description to ensure clarity for the reader. To this end, all beam splitters and polarization analyser ensembles have been numbered. Its caption was shortened whereas section 4.2 was detailed.

The Bertrand lens images the back focal plane of the collection lens. The Fourier image is projected at infinity and the tube lens is used to create the image on the camera. This configuration, discussed in *J. Opt. Soc. Am. A* **32**, 2082-2092 (2015), enables us to move from real to Fourier plane by adding one optical element along the detection path.

The beam-splitter BS3 is a non-polarizing 10:90 BS that splits both the SH sample and reference beams. Actually, BS3 ensures spatial overlap between 10% of the reference beam that is reflected by BS3 and 90% of the sample beam that is transmitted by BS3. This, together with the temporal overlap provided by the delay line, gives us the means to characterize the phase of the SH signal emitted by the AlGaAs thin film, unraveling the phase singularity of the OAM carrying sample beam in the inset of Fig 2(c). The transmitted reference and reflected sample beams at BS3 are not analyzed and blocked by a beam dump.

The polarization analyzer in the transmission path of the microscope (QP2-LP2) allows the characterization of the polarization properties of the transmitted SH beam, as shown on Fig 2(c) and 2(g). Similarly, the polarization analyzer in the reflection path of the microscope (QP3+LP3) gives us the means to characterize the polarization properties of the reflected SH beam as shown on Fig(d) and (h).

The experimental setup section has been improved as follows: “The nonlinear characterization setup is based on a horizontal microscope presented on Fig. 5. The pump field is generated by an optical parametric amplifier (OPA - Mango, APE) pumped by a mode-locked Yb-doped fiber laser (Satsuma, Amplitude). This source generates 160 fs-pulses at 1550 nm with a repetition rate of 500 kHz.

The pump beam size is controlled by a telescope, its polarization state is set by a linear polarizer and a quarter-wave plate and the pump OAM is tuned by reflecting the beam on a SLM (SLM-200-21, Santec).

Precise control of the pump power is achieved using a half-wave plate and a Glan-Taylor polarizer, it is then measured by sending a fraction of this power on an InGaAs photodiode through beam-splitter BS2.

The pump beam is focused on the sample by a micro-lens with NA=0.16 and high transmission at the fundamental frequency. The reflected (resp. transmitted) pump beam is removed by a short-pass dichroic mirror (resp. short-pass filter). After being collected by a micro-lens (NA=0.68), the forward "transmitted" SH signal is sent either on a CCD camera (Sony, ICX825AL), a spectrometer (Broadcom, Qmini), or a Si photodiode (Newport, 818-SL). We move from real to Fourier space by imaging the back focal plane of the collection lens with a Bertrand lens.

For phase imaging of the SH signal, we let it interfere with a reference beam at SH in a Mach-Zender interferometer that includes beam splitters BS1 and BS3. The former sends a part of the pump beam to the reference path, where first a phase-matched BBO crystal doubles its frequency; then a delay line, consisting of a micrometric precision motorized stage, ensures the temporal overlap of the two pulses on BS3. The resulting interference pattern is recorded on the CCD, while the transmitted reference and reflected sample beams at BS3 are sent to a beam dump (BD).”

Reviewer 2: *2- The results presented in the manuscript clearly demonstrate that the angular momentum is not conserved in the process, which is natural due to the anisotropy of the nonlinear medium. The authors should comment on that and make a quick estimate of the torque exerted on the film. As a thin layer, maybe it is not completely impossible to devise a measurement scheme in a future work.*

Authors' reply: We agree with the Reviewer that a torque occurs in our system due to non-conservation of total angular momentum in second-harmonic generation. An estimate of the torque can be obtained by following the procedure outlined in Toftul et al., Phys. Rev. Lett. 130, 243802, 2023 for nonlinearity-induced optical torque of small objects and adapted to the case of thin films. The maximum torque exerted from light with intensity I_0 on an object with geometric cross-section σ_{geo} is $T_0 = I_0 \sigma_{geo} n / \omega$, where n is the object refractive index and ω the angular frequency of light. If the object is transparent at the fundamental frequency and the dominant nonlinearity is quadratic, the torque originates from the second-harmonic generation process. We first evaluate the conversion efficiency for second-harmonic generation in reflection and transmission. Let's assume that the pump has spin equal to +1 and 0 topological charge (circularly-polarized gaussian beam). The incident power is then $P_{inc} = I_0 \pi w^2 / 2$. The second-harmonic power in reflection, given by the integration in the xy plane of the power density carried by the LG mode with spin -1 and charge +1 generated in reflection is $P_r = I_0 A^2 [\chi_S^{(2)}]^2 \pi / 8$, where $A = \sqrt{2 \eta_0 I_0}$ is the amplitude of the input electric field, η_0 the vacuum impedance and $\chi_S^{(2)}$ is the surface susceptibility of the nonlinear sheet ($\chi_S^{(2)} = \chi^{(2)} d$). Therefore, the efficiency in reflection is $P_r / P_{inc} = \frac{A^2 [\chi_S^{(2)}]^2}{4 w^2}$. The efficiency is of the order of 10^{-9} for a 100-nm-thick film of AlGaAs ($\chi^{(2)}=200$ pm/V) illuminated by a gaussian beam with peak intensity of $I_0 = 1$ GW/cm² and beam waist $w = 30$ μ m. In transmission, the efficiency is due to two LG modes: one of them has the same intensity of the reflected one, and the other has quadruple intensity. Hence, the efficiency in transmission, P_t / P_{inc} , is five times larger than the efficiency in reflection. The optomechanical torque associated with the absorption of two pump photons with total angular momentum equal to $m_{inc} = 2(l^\omega + \sigma^\omega) = 2$ that generate one second-

harmonic photon is $T_l = m_{inc} T_0 \frac{P_r + P_t}{P_{inc}}$. The torque associated with the angular momentum of the emitted second-harmonic light should be added to T_l . However, for the process under investigation, described in Fig. 1 of the main text, second-harmonic light is emitted with zero net angular momentum, both in reflection and in transmission. Therefore, considering that the maximum torque for a gaussian beam at $\lambda = 1.55 \mu\text{m}$ on an AlGaAs film ($n \approx 3.5$) is $T_0 = P_{inc} n / \omega \approx 4 \times 10^{-11} \text{ N}\cdot\text{m}$, the total torque is $T_l = T_0 \frac{3A^2[\chi_s^{(2)}]^2}{w^2} \approx 4 \times 10^{-19} \text{ N}\cdot\text{m}$. The torque may be larger and, in some cases, negative [see Phys. Rev. Lett. 130, 243802, 2023], when the pump carries orbital angular momentum in addition to spin angular momentum.

Reviewer 2: 3- *The authors should also consider the use of vector beams (radially or azimuthally polarized) since their longitudinal electric field component is increased as compared to the usual scalar beams.*

Authors' reply: We think this is a very interesting suggestion and it certainly deserves an adequate investigation. We are currently planning to start a theoretical and experimental work focused on spin-orbit interactions involving radially and azimuthally polarized vector beams.

Reviewer 2: 4- *I do not agree with the terminology used for the quantities calculated in Eqs. (23) and (24). There is a literature about fractional angular momentum, which is related to different kinds of phase and polarization structures. For example, the authors should look at PRL 95, 240501 (2005) and Sci. Adv. 9, eadf3486 (2023). The quantity calculated by the authors is more related to the average spin and angular momenta since they come from the weighted sum of these quantities in the transmitted light.*

Authors' reply: We acknowledge that the word fractional is usually referred to other types of beams and phase structures, and therefore we have accepted the Reviewer's suggestion and kept only the word "average" throughout the manuscript, in reference to the quantities in Eqs. (23) and (24).

List of other minor modifications and corrections

- Eq. 13 has been modified by inserting a factor $1/\sqrt{2}$ to normalize the pump field in a way fully consistent with Eq. 1.

- Fig. 2: after reexamining the data for the revised version, we had a minor doubt about the alignment of the setup and preferred to redo and replot the experiments of Fig. 2a-2d to lift any ambiguity about this point. The result is essentially the same as the original measurements.
- The second to last sentence of the introduction has been modified as follows: “Then we report experimental results of SHG from a thin film of (001) AlGaAs on a transparent substrate excited with pump beams carrying different combinations of SAM and OAM, showing spin-orbit coupling effects in full agreement with our theoretical predictions”
- At the end of section 2.2, the following words have been canceled: “associated with an average value of SAM and OAM”
- Last line of the abstract: "when pumped with a circularly" replaced with "even when pumped with a simple circularly”.

Reviewer #2 (Remarks to the Author):

The authors have properly dealt with the points raised in my first report and improved considerably both the discussion and the experimental results. Now, they included experimental results with $l_w \neq 0$, giving more strength to the central claim of the work. In doing so, the authors observed the appearance of a radial mode when $l_w \cdot \sigma_w < 0$. In this regard, the authors should comment on previous works showing the appearance of radial modes caused by radial-angular coupling and chiral relations in nonlinear wave mixing:

- PHYSICAL REVIEW A 96, 053856 (2017)
- PHYSICAL REVIEW A 101, 043821 (2020)
- PHYSICAL REVIEW APPLIED 16, 044019 (2021)
- PHYSICAL REVIEW A 108, 013503 (2023)

In these works, the appearance of radial modes was explained in terms of the selection rules that arise from the spatial overlap between the interacting modes. Once this minor point is addressed, the manuscript can be accepted for publication in Nature Communications. The demonstration of angular momentum transfer in nonlinear wave mixing in thin films is an important step towards the development of technological devices for spin-orbit manipulation of light beams.

Manuscript NCOMMS-23-43699A Response to the Reviewer's comment

Reviewer #2:

The authors have properly dealt with the points raised in my first report and improved considerably both the discussion and the experimental results. Now, they included experimental results with $l_w \neq 0$, giving more strength to the central claim of the work. In doing so, the authors observed the appearance of a radial mode when $l_w \cdot \sigma_w < 0$. In this regard, the authors should comment on previous works showing the appearance of radial modes caused by radial-angular coupling and chiral relations in nonlinear wave mixing:

- PHYSICAL REVIEW A 96, 053856 (2017)
- PHYSICAL REVIEW A 101, 043821 (2020)
- PHYSICAL REVIEW APPLIED 16, 044019 (2021)
- PHYSICAL REVIEW A 108, 013503 (2023)

In these works, the appearance of radial modes was explained in terms of the selection rules that arise from the spatial overlap between the interacting modes. Once this minor point is addressed, the manuscript can be accepted for publication in Nature Communications. The demonstration of angular momentum transfer in nonlinear wave mixing in thin films is an important step towards the development of technological devices for spin-orbit manipulation of light beams.

Authors' reply:

We thank the Reviewer for recognizing that the manuscript has been considerably improved and for the comment about the appearance of higher radial order in SH generation. As reported by the Reviewer, a similar effect was observed in type II second-harmonic generation with orbital-angular-momentum mixing of counter-rotating vortices [PHYSICAL REVIEW A 96, 053856 (2017)], in spontaneous-parametric down conversion [PHYSICAL REVIEW APPLIED 16, 044019 (2021)] and in self-phase modulation [PHYSICAL REVIEW A 108, 013503 (2023)], and interpreted as a diffraction effect in [PHYSICAL REVIEW A 101, 043821 (2020)].

The peculiarity of nonlinear spin-orbit coupling in AlGaAs is that the appearance of a higher radial order is related to the presence of spin in the pump beam, as well as to the $\chi^{(2)}$ tensor symmetry of AlGaAs, which produces two LG beams in transmission and one in reflection, with topological charge that can be generally written as $l^{2\omega} = 2l^\omega \pm \sigma^\omega$. As discussed in the manuscript, among the three LG beams, only those for which the absolute value of the topological charge decreases with respect to the conventional topological-charge-doubling rule, i.e., those with $|l^{2\omega}| = 2|l^\omega| - 1$, acquire an increased radial index (equal to 1).

We have added the references indicated by the Reviewer and the following sentence in the Results section, after Eq. 22: "The appearance of a higher radial order has been observed in type II second-harmonic generation with orbital-angular-momentum mixing of counter-rotating vortices [26], in spontaneous-parametric down conversion [27] and in self-phase modulation [28], and it has been interpreted as a diffraction effect in [29]. The peculiarity of nonlinear spin-orbit coupling in AlGaAs is that the appearance of a higher radial order is strictly related to the presence of spin in the pump beam, as well as to the $\chi^{(2)}$ tensor symmetry".